# Hearing impairment among adult foreign-born and Swedish-born individuals: A national Swedish study

Per Wändell[1,2], Xinjun Li[2], Axel Carlsson [1,3] *, Jan Sundquist[2,4,5], Kristina Sundquist[2,4,5]

**1** Division of Family Medicine and Primary Care, Department of Neurobiology, Care Sciences and Society, Karolinska Institutet, Huddinge, Sweden, **2** Center for Primary Health Care Research, Lund University, Malmö, Sweden, **3** Academic Primary Health Care Centre, Stockholm Region, Stockholm, Sweden, **4** Department of Family Medicine and Community Health, Department of Population Health Science and Policy, Icahn School of Medicine at Mount Sinai, New York, United States of America, **5** Department of Functional Pathology, Center for Community-Based Healthcare Research and Education (CoHRE), School of Medicine, Shimane University, Matsue, Japan

* axel.carlsson@ki.se

**Data Availability Statement:** All relevant data are within the article and its Supporting Information files.

**Funding:** The authors received no specific funding for this work.

## Abstract

### Objectives

To analyze the risk of hearing impairment in adult first-generation immigrants, i.e., foreign-born individuals as compared to Swedish-born individuals.

### Study design

A register-based study follow-up study.

### Methods

A nationwide study of individuals 25 years of age and older (N = 5 464 245; 2 627 364 men and 2 836 881 women) in Sweden. Hearing impairment was defined as at least one registered diagnosis in the National Patient Register between January 1st, 1998 and December 31st, 2015. Cox regression analysis was used to estimate the relative risk (hazard ratios (HR) with 95% confidence intervals (CI)) of incident hearing impairment in foreign-born compared to Swedish-born individuals. Cox regression models were stratified by sex and adjusted for age, comorbidities, and socioeconomic status.

### Results

A total of 244 171 cases (124 349 men and 119 822 women) of hearing impairment were registered. Hearing impairment risk expressed as fully adjusted HRs (99% CI) was somewhat lower among immigrant men 0.95 (0.92–97) but not among immigrant women 0.97 (0.95–1.00), with significantly higher fully adjusted HRs among men and women from Asia, and Eastern Europe, and women from Africa.

**Competing interests:** The authors have declared
that no competing interests exist

## Conclusions

We observed a somewhat lower risk of hearing impairment among foreign-born men, but
there was a higher risk among men and women from some regions.

## 1. Introduction

Globally, hearing loss is estimated to be "the fourth most leading contributor to years lived
with disability" [1]. The causes of deafness and hearing impairment differ by world region.
Furthermore, the risk of hearing impairment also varies in different regions of the world, with
the lowest risks found in high income countries, i.e. mostly Western and Northern Europe and
Northern America [2]. By contrast, higher risks are found in Central and Eastern Europe, Sub-
Saharan Africa, Latin America, and Asian regions, mostly south Asia. Globally, half a billion
people are estimated to exhibit a disabling hearing loss, i.e. around 7% of the world's popula-
tion [1, 2]. An American study found that the prevalence of deafness or serious difficulty hear-
ing was approximately 6% [3].

Several comorbidities have been associated with hearing loss including visual impairment,
mobility restrictions, psychosocial health problems, diabetes, cardiovascular diseases, stroke,
arthritis, cancer [4], dementia, depression, and falls [5]. Lower socioeconomic status is often
associated with a higher risk of several diseases, both at the individual [6] and neighborhood
level [7]. Lower socioeconomic status has also been found to be associated with hearing
impairment [3]. Aside from lower socioeconomic status, not being married is also a risk factor
for several morbidities and higher mortality, especially among men [8], and may therefore also
be associated with hearing loss.

As regards immigrants, hearing impairment may be challenging when learning the language
and culture of a new country, which calls for attention and motivated us to conduct this study.
Identifying hearing impairment is also important as correction with hearing aids and rehabilitative
services have been shown to increase well-being and quality of life [9]. An American study found
differences between Asian groups of elderly foreign-born people, with a higher risk of deafness
among Hawaiians/Pacific Islanders, Filipinos, and non-Hispanic whites than among Chinese
immigrants [10]. Sweden is a country with a high proportion of both first- and second-generation
immigrants but we have not been able to find any previous Swedish studies on hearing impairment
among foreign-born individuals (i.e., first-generation immigrants) that were conducted in the
country. We have previously shown that, among second-generation immigrants, boys with parents
from Asia have a higher risk of extended sensorineural hearing impairment [11].

The aim of this study was to compare the risk of hearing impairment in immigrant men
and women with Swedish-born men and women. Based on earlier findings [2], we hypothe-
sized that some immigrant groups, including non-Western regions and Central and Eastern
Europe, may have an increased risk of hearing impairment, which is important to investigate
as it may affect immigrants' integration in Sweden.

## 2. Methods

### 2.1 Design

We used the Swedish Total Population Register and the National Patient Register (NPR) for
the study. The Total Population Register, which is based on unique personal identification
numbers, includes data on all individuals registered in Sweden, [12, 13]. Individuals aged 25

years of age and older were included to compare foreign-born individuals to Swedish-born. The follow-up period ran from January 1st, 1998 until hospitalization/out-patient treatment of a diagnosis of hearing impairment, death, emigration, or the end of the study period on December 31st, 2015, whichever, came first. Nationwide out-patient diagnoses were included from 2001 onwards from specialist open care but not primary health care as these diagnoses are not included in the NPR. The unique pseudonymized serial number for each individual was used to avoid double-counting.

## 2.2 Outcome variable

Hearing loss based on ICD-10 codes included the following conditions: Conductive and sensorineural hearing loss (H90), other hearing loss (H91), and noise-induced hearing loss (H83.3). In a categorization of different diagnoses of hearing loss we included the following subgroups: "Conductive hearing impairment" (H90.0-H90.2); "Extended sensorineural hearing impairment" (H90.3-H90.5), also including ototoxic hearing loss (H91.0), presbycusis (H91.1); "Noise-induced hearing impairment" (H83.3); and finally "Other hearing impairment" (H91.2-H91.9), also including mixed etiology (H90.6-H90.8).

## 2.3 Comorbidities

We identified the following comorbidities (with ICD-10 codes) based on the existing literature [4, 14]: Tinnitus (H93.1); Intracranial trauma (S06); Malignant brain tumor (D32, D33, C70, C71); Stroke (I60-I69); Hypertension (I10-I19); Coronary Heart Disease (CHD; I20-I25); Chronic Obstructive Pulmonary Disease (COPD; J40-J47); Cancer (C00-C97, with C70 and C71 excluded, and categorized into the brain tumor group); Diabetes (E10-E14); Arthropathies (M00-M25, also including Arthrosis M15-M19); Dementia (F00-F03, F10.7A, G30, G31.8A); Depression (F32-F33); and Visual impairment (H54).

## 2.4 Demographic and socioeconomic variables

Age was used as a continuous variable in the analysis.

Marital status was defined as married or not.

Educational attainment was categorized as ≤9 years (partial or complete compulsory schooling), 10–12 years (partial or complete secondary schooling) and >12 years (attendance at college and/or university).

Geographic region of residence was included in order to adjust for possible regional differences in health care access and was categorized as (1) large cities with surrounding regions, (2) southern Sweden (southern and middle part of Sweden) and (3) northern Sweden (the five most northern counties). Regarding regions in Sweden, the urbanicity differs with many sparsely populated parts in northern Sweden with, in many cases, long transportation routes and also poorer access to ophthalmologists. Large cities were defined as municipalities with a population of >200,000 and comprised the three largest cities in Sweden: Stockholm, Gothenburg and Malmö.

The analyses were stratified by sex because men and women experience different types of environments, including those related to occupation, and also have different health care seeking patterns [15].

## 2.5 Neighborhood deprivation

Neighborhood socioeconomic status (NSES) has been shown to be an important socioeconomic factor for several health outcomes. The NSES was derived from Small Area Market

Statistics (SAMS). The neighborhoods were derived from Small Area Market Statistics (SAMS), which were originally created for commercial purposes and pertain to small geographic areas with boundaries defined by homogenous types of buildings. The average population in each SAMS neighborhood is approximately 2000 people for Stockholm and 1000 for the rest of Sweden. A summary index was calculated to characterize neighborhood-level deprivation. The neighborhood index was based on information about female and male residents, aged 20 to 64 years of age, because this age group represents those who are among the most socioeconomically active in the population (i.e. a group that has a stronger impact on the socioeconomic structure in the neighborhood compared to children, younger women and men, and retirees). The index was based on the following four variables: low educational status (<10 years of formal education); income from all sources, including interest and dividends, that is (<50% of the median individual income); unemployment (excluding full-time students, those completing military service, and early retirees); and receipt of social welfare [6, 7]. This index was categorized into four groups: more than one standard deviation (SD) below the mean (low deprivation level or high SES), more than one SD above the mean (high deprivation level or low SES), and within one SD of the mean (moderate SES or moderate deprivation level) used as the reference group, and also unknown neighborhood SES.

## 2.6 Statistical analysis

Baseline data are presented with continuous variables as mean and standard deviations and categorical variables as counts and percentages. We used Cox regression analysis to estimate the relative risk (hazard ratios (HR) with 95% confidence intervals (CI)) of incident hearing impairment in separate groups of foreign-born individuals compared to the control group, i.e. Swedish-born, during the follow-up time. We used an open cohort design and as Cox regression was used in the statistical analysis, only the first event was registered. Risk time was calculated until the event and those who died or emigrated were censored. All analyses were stratified by sex. Three models were used: Model 1 with adjustment for age and region of residence; Model 2 with adjustment for age, region of residence, educational level, marital status and neighborhood SES; this was to examine to what extent SES explained the association between country of birth and incident hearing impairment; and Model 3 as Model 2 but with the inclusion also of relevant comorbidities (tinnitus, intracranial trauma, malignant brain tumor, stroke, CHD, COPD, cancer, diabetes, arthropathies, dementia, depression, and visual impairment); this was to examine if other diagnoses explained the association between country of birth and hearing impairment.

We also sub-divided into groups of hearing impairment, i.e. into "Conductive hearing impairment", "Extended sensorineural hearing impairment", "Noise-induced hearing impairment", and "Other hearing impairment".

In a sensitivity analysis, we adjusted for time in Sweden for immigrants. In addition, we also analyzed the number of separate diagnoses for men and women.

## 2.7 Ethical approval

All procedures performed in studies involving human participants were in accordance with the ethical standards of the institutional and/or national research committee and with the 1964 Helsinki declaration and its later amendments or comparable ethical standards.

Informed consent was not applicable, as the study was based on pseudonymized data from registers.

The study was approved by the Regional Ethical Review Board in Lund.

The authors are not allowed to share the used data from the data sources being used due to legal restrictions in Sweden.

## 3. Results

### 3.1 Main results

In total (Table 1), 5 464 245 individuals were included (2 627 364 men and 2 836 881 women), of which 244 171 individuals had a registered diagnosis of hearing impairment (124 349 men and 119 822 women).

Of the included 2 627 364 men, 2 193 544 were Swedish-born, of which 108 788 men had hearing impairment (mean age 64.1 years, SD 12.8). Among the 433 820 foreign-born men, 15 561 had hearing impairment (mean age 59.0 years, SD 13.0; S1a Table in S1 File). Among the women, 2 409 298 were Swedish-born, of which 105 260 had hearing impairment (mean age 65.7 years, SD 14.1), and 427 583 were foreign-born of which 14 562 had hearing impairment (mean age 60.7 years, SD 14.2; S1b Table in S1 File). The incidence rate was higher among foreign-born persons up to 60 years of age, and higher among Swedish-born individuals 60 years of age and above (data not shown). Of the included comorbid conditions, most were associated with a higher risk of hearing impairment (S2a and S2b Tables in S1 File). The most common comorbidities are shown in Table 1. Regarding sociodemographic factors, a higher educational level was associated with a slightly higher risk especially among foreign-born persons (S2a and S2b Tables in S1 File). Being married was also associated with a higher risk among men and among Swedish-born women, while neighborhood socioeconomic status showed no significantly different risk or was only marginally different (except for unknown level with a lower risk among Swedish-born men and women). Mean time to event was for men 14.2 years (SD 4.9) and women 14.5 years (SD 4.7); for Swedish-born men 14.3 years (SD 4.9) and foreign-born men 13.6 years (SD 5.3); and for Swedish-born women 14.5 years (SD 4.7) and foreign-born women 14.3 years (SD 4.8).

### 3.2 Differences between foreign-born and Swedish-born individuals

The results for foreign-born men and women compared to Swedish-born men and women (with the number of included individuals and cases in S3 Table in S1 File), respectively, are shown in Table 2, with slightly lower risks among foreign-born men but not among foreign-born women in the fully adjusted models. The risk was consistently higher in men and women from Asia, and also higher in models 2 and 3 in men and women from Eastern Europe, and women from Africa (Table 2). Results for specific countries (S4a and S4b Tables in S1 File) showed, for Asian countries, a consistently higher risk that was observed for both men and women from Turkey, Lebanon, Iran, and Iraq, and also, in models 2 and 3, in men and women from other Asian countries. For European countries, a consistently higher risk was found in men from Bosnia, and, in models 2 and 3, in women from Bosnia and former Yugoslavia. Lower risks were consistently found for men and women from all the Nordic countries (Denmark, Finland, Iceland, and Norway), and for most Southern European countries (i.e. for men from Greece, Italy and Spain, and women from France, Greece, and Italy), and also among men and women from UK and Ireland, and men from Hungary. In a sensitivity analysis, we also adjusted for time in Sweden (S5 Table in S1 File), with only minor differences from the earlier shown results, i.e. results were non-significant for men from Eastern Europe (HR 0.97, 99% CI 0.91–1.04), but significantly lower for men from Africa (HR 0.84, 99% CI 0.74–0.95) and Latin America (HR 0.87, 99% CI 0.87–0.99).

Categorization into different types of hearing impairment is shown in Table 3. Compared to Swedish-born individuals, the risk of conductive hearing impairment was higher in men

**Table 1. Study population and number of cases with hearing impairment categorized by sex.**

| | Men | | | | Women | | | |
|---|---|---|---|---|---|---|---|---|
| | Whole Population | | Hearing impairment | | Whole Population | | Hearing impairment | |
| | No. | % | No | % | No. | % | No | % |
| Total population | 2627364 | | 124349 | | 2836881 | | 119822 | |
| Age (years) | | | | | | | | |
| 25–39 | 811202 | 30.9 | 16929 | 13.6 | 836038 | 29.5 | 16449 | 13.7 |
| 40–49 | 513920 | 19.6 | 23376 | 18.8 | 527349 | 18.6 | 21001 | 17.5 |
| 50–59 | 532459 | 20.3 | 38405 | 30.9 | 535011 | 18.9 | 31143 | 26.0 |
| $\geq 60$ | 769783 | 29.3 | 45639 | 36.7 | 938483 | 33.1 | 51229 | 42.8 |
| Educational level | | | | | | | | |
| $\leq 9$ | 887937 | 33.8 | 41982 | 33.8 | 991397 | 34.9 | 41592 | 34.7 |
| 10–12 | 712203 | 27.1 | 33223 | 26.7 | 815714 | 28.8 | 38872 | 32.4 |
| > 12 | 1027224 | 39.1 | 49144 | 39.5 | 1029770 | 36.3 | 39358 | 32.8 |
| Region of residence | | | | | | | | |
| Large cities | 864284 | 32.9 | 38724 | 31.1 | 960205 | 33.8 | 41418 | 34.6 |
| Southern Sweden | 1104889 | 42.1 | 58902 | 47.4 | 1213259 | 42.8 | 55323 | 46.2 |
| Northern Sweden | 658191 | 25.1 | 26723 | 21.5 | 663417 | 23.4 | 23081 | 19.3 |
| Marital status | | | | | | | | |
| Married | 1576047 | 60.0 | 89548 | 72.0 | 2535098 | 89.4 | 107755 | 89.9 |
| Not married | 1051317 | 40.0 | 34801 | 28.0 | 301783 | 10.6 | 12067 | 10.1 |
| Neighborhood deprivation | | | | | | | | |
| Low | 366266 | 13.9 | 20069 | 16.1 | 392916 | 13.9 | 18558 | 15.5 |
| Middle | 1249190 | 47.5 | 64509 | 51.9 | 1390177 | 49.0 | 61238 | 51.1 |
| High | 296743 | 11.3 | 13660 | 11.0 | 329470 | 11.6 | 13270 | 11.1 |
| Unknown | 715165 | 27.2 | 26111 | 21.0 | 724318 | 25.5 | 26756 | 22.3 |
| Hospital diagnoses: | | | | | | | | |
| Tinnitus | 36126 | 1.4 | 21854 | 17.6 | 35658 | 1.3 | 17920 | 15.0 |
| Intracranial trauma | 70954 | 2.7 | 4986 | 4.0 | 63075 | 2.2 | 4075 | 3.4 |
| Brain tumor | 15322 | 0.6 | 2049 | 1.6 | 21491 | 0.8 | 2490 | 2.1 |
| Stroke | 257571 | 9.8 | 16311 | 13.1 | 252308 | 8.9 | 13408 | 11.2 |
| Hypertension | 477208 | 18.2 | 37544 | 30.2 | 524806 | 18.5 | 36384 | 30.4 |
| CHD | 352757 | 13.4 | 26508 | 21.3 | 250013 | 8.8 | 16496 | 13.8 |
| COPD | 150019 | 5.7 | 10169 | 8.2 | 190049 | 6.7 | 11492 | 9.6 |
| Cancer (except brain tumors) | 459133 | 17.5 | 33370 | 26.8 | 464865 | 16.4 | 27245 | 22.7 |
| Diabetes | 222195 | 8.5 | 14890 | 12.0 | 181279 | 6.4 | 10719 | 8.9 |
| Arthropathy | 481502 | 18.3 | 35438 | 28.5 | 620217 | 21.9 | 40378 | 33.7 |
| Dementia | 70809 | 2.7 | 4576 | 3.7 | 102585 | 3.6 | 5221 | 4.4 |
| Depression | 86003 | 3.3 | 5571 | 4.5 | 138001 | 4.9 | 7786 | 6.5 |
| Visual impairment | 6248 | 0.2 | 632 | 0.5 | 7880 | 0.3 | 725 | 0.6 |

CHD: Coronary heart disease; COPD: Chronic obstructive pulmonary disease.

and women from Eastern Europe, Africa and Asia, the risk of extended sensorineural hearing impairment was higher in men and women from Asia, the risk of noise-induced hearing impairment was higher in women from Eastern Europe (although the cases were few), and, finally, the risk of other types of hearing impairment was higher in men and women from Eastern Europe and Asia, and also in women from Africa and Russia.

**Table 2. Relative risk of hearing impairment in foreign-born men and women with Swedish-born men and women as referents, respectively, expressed as hazard ratios (HR) with 99% confidence intervals (99% CI).**

| | Obs. | Model 1 | | | Model 2 | | | Model 3 | | |
|---|---|---|---|---|---|---|---|---|---|---|
| | | HR | 99% CI | | HR | 99% CI | | HR | 99% CI | |
| **Men** | | | | | | | | | | |
| Sweden | 108788 | 1 | | | 1 | | | 1 | | |
| **All foreign-born men** | 15 561 | **0.83** | **0.81** | **0.85** | **0.94** | **0.92** | **0.97** | **0.95** | **0.92** | **0.97** |
| **Nordic countries** | 4745 | **0.68** | **0.65** | **0.71** | **0.79** | **0.75** | **0.82** | **0.82** | **0.79** | **0.86** |
| **Southern Europe** | 637 | **0.53** | **0.47** | **0.59** | **0.64** | **0.57** | **0.72** | **0.68** | **0.60** | **0.76** |
| **Western Europe** | 1288 | **0.73** | **0.67** | **0.79** | **0.81** | **0.74** | **0.87** | **0.84** | **0.78** | **0.91** |
| **Eastern Europe** | 2204 | 1.01 | 0.95 | 1.07 | **1.14** | **1.07** | **1.21** | **1.08** | **1.01** | **1.15** |
| **Baltic countries** | 242 | 0.91 | 0.76 | 1.10 | 0.95 | 0.79 | 1.14 | 0.98 | 0.82 | 1.18 |
| **Central Europe** | 763 | **0.81** | **0.73** | **0.90** | **0.83** | **0.75** | **0.92** | **0.84** | **0.76** | **0.93** |
| **Africa** | 590 | **0.76** | **0.68** | **0.86** | 0.93 | 0.83 | 1.05 | 0.93 | 0.83 | 1.05 |
| **Northern America** | 224 | **0.58** | **0.48** | **0.70** | **0.69** | **0.57** | **0.83** | **0.76** | **0.63** | **0.92** |
| **Latin America** | 532 | **0.82** | **0.73** | **0.93** | 0.95 | 0.84 | 1.08 | 0.97 | 0.86 | 1.10 |
| **Asia** | 4169 | **1.26** | **1.20** | **1.31** | **1.45** | **1.39** | **1.52** | **1.33** | **1.27** | **1.39** |
| **Russia** | 111 | **0.74** | **0.57** | **0.97** | 0.85 | 0.65 | 1.11 | 0.86 | 0.66 | 1.13 |
| **Women** | | | | | | | | | | |
| Sweden | 105260 | 1 | | | 1 | | | 1 | | |
| **All foreign-born women** | 14 562 | **0.89** | **0.87** | **0.91** | 1.00 | 0.97 | 1.03 | 0.97 | 0.95 | 1.00 |
| **Nordic countries** | 6203 | **0.81** | **0.78** | **0.84** | **0.90** | **0.86** | **0.93** | **0.91** | **0.87** | **0.94** |
| **Southern Europe** | 311 | **0.48** | **0.40** | **0.56** | **0.58** | **0.49** | **0.68** | **0.62** | **0.52** | **0.72** |
| **Western Europe** | 1248 | **0.89** | **0.82** | **0.97** | 0.94 | 0.86 | 1.02 | 0.93 | 0.86 | 1.01 |
| **Eastern Europe** | 1488 | 0.95 | 0.88 | 1.02 | **1.13** | **1.05** | **1.22** | **1.13** | **1.05** | **1.22** |
| **Baltic countries** | 241 | 0.85 | 0.70 | 1.02 | 0.85 | 0.71 | 1.02 | 0.85 | 0.71 | 1.03 |
| **Central Europe** | 919 | 0.93 | 0.84 | 1.02 | 0.94 | 0.86 | 1.04 | 0.94 | 0.85 | 1.03 |
| **Africa** | 379 | 1.04 | 0.89 | 1.20 | **1.32** | **1.14** | **1.53** | **1.32** | **1.14** | **1.53** |
| **Northern America** | 172 | **0.58** | **0.47** | **0.72** | **0.67** | **0.54** | **0.83** | **0.68** | **0.55** | **0.85** |
| **Latin America** | 498 | 0.95 | 0.84 | 1.08 | 1.07 | 0.94 | 1.22 | 1.08 | 0.95 | 1.22 |
| **Asia** | 2874 | **1.27** | **1.20** | **1.34** | **1.55** | **1.47** | **1.64** | **1.53** | **1.44** | **1.62** |
| **Russia** | 177 | 0.84 | 0.68 | 1.04 | 0.96 | 0.77 | 1.19 | 0.96 | 0.78 | 1.19 |

Model 1: adjusted for age and region of residence in Sweden; model 2: adjusted for age, region of residence in Sweden, educational level, and marital status, and neighborhood deprivation; model 3: model 2 + comorbidities (tinnitus, intracranial trauma, malignant brain tumor, stroke, CHD, COPD, cancer, diabetes, arthropathies, dementia, depression, and visual impairment).

Bold values are statistically significant.

### 3.3 Sensitivity analyses

We analyzed the number of all diagnoses of hearing impairments in men and women, respectively (S6a and S6b Tables in S1 File). The most common diagnoses were bilateral sensorineural hearing loss (ICD-10 code H90.3), which was found among men (38.9%) and among women (36.6%), and unspecified sensorineural hearing loss (ICD-10 code H90.5), among men 25.0% and among women 23.9%. Presbyacusis (H91.1) was registered only among 6.05 among men and 7.8% among women.

## 4. Discussion

The main results of this study were that the overall risk of hearing impairment among immigrant men in Sweden was somewhat lower than among Swedish-born men. This was, however,

**Table 3. Relative risks of conductive hearing loss and other types of hearing loss in foreign-born men and women with Swedish-born men and women as referents, respectively, expressed as hazard ratios (HR) with 99% confidence intervals (99% CI)\*.**

| | Conductive hearing impairment | | | | Extended sensorineural hearing impairment | | | | Noise-induced hearing impairment | | | | Other hearing impairment | | | |
|---|---|---|---|---|---|---|---|---|---|---|---|---|---|---|---|---|
| | Obs. | HR | 99% CI | | Obs. | HR | 99% CI | | Obs. | HR | 99% CI | | Obs. | HR | 99% CI | |
| **Men** | | | | | | | | | | | | | | | | |
| Sweden | 4669 | 1 | | | 85335 | 1 | | | 3289 | 1 | | | 15495 | 1 | | |
| Nordic countries | 255 | 1.02 | 0.84 | 1.22 | 3617 | 0.81 | 0.77 | 0.85 | 156 | 0.94 | 0.74 | 1.19 | 717 | 0.86 | 0.77 | 0.96 |
| Southern Europe | 38 | 0.94 | 0.59 | 1.49 | 459 | 0.63 | 0.55 | 0.72 | 14 | 0.54 | 0.25 | 1.16 | 126 | 0.90 | 0.70 | 1.16 |
| Western Europe | 47 | 0.73 | 0.48 | 1.12 | 996 | 0.83 | 0.76 | 0.91 | 20 | 0.56 | 0.29 | 1.05 | 225 | 0.99 | 0.81 | 1.20 |
| Eastern Europe | 205 | **1.91** | **1.54** | **2.37** | 1494 | 0.95 | 0.88 | 1.03 | 74 | 1.29 | 0.92 | 1.82 | 431 | **1.46** | **1.27** | **1.68** |
| Baltic countries | 4 | 0.54 | 0.13 | 2.26 | 190 | 0.96 | 0.78 | 1.19 | 4 | 0.88 | 0.21 | 3.67 | 44 | 1.17 | 0.76 | 1.79 |
| Central Europe | 38 | 1.00 | 0.63 | 1.59 | 551 | 0.78 | 0.69 | 0.88 | 13 | 0.58 | 0.26 | 1.29 | 161 | 1.15 | 0.92 | 1.45 |
| Africa | 99 | **2.46** | **1.83** | **3.32** | 359 | 0.75 | 0.65 | 0.87 | 11 | 0.57 | 0.24 | 1.36 | 121 | 1.25 | 0.96 | 1.63 |
| North America | 7 | 0.50 | 0.17 | 1.47 | 164 | 0.72 | 0.57 | 0.89 | 10 | 1.44 | 0.58 | 3.57 | 43 | 0.98 | 0.63 | 1.51 |
| Latin America | 45 | 1.49 | 0.97 | 2.29 | 374 | 0.91 | 0.78 | 1.05 | 12 | 0.68 | 0.30 | 1.55 | 101 | 1.19 | 0.89 | 1.58 |
| Asia | 480 | **2.61** | **2.24** | **3.03** | 2795 | **1.17** | **1.11** | **1.24** | 103 | 0.99 | 0.73 | 1.33 | 791 | **1.75** | **1.56** | **1.95** |
| Russia | 4 | 0.67 | 0.16 | 2.78 | 78 | 0.77 | 0.56 | 1.07 | 2 | 0.70 | 0.09 | 5.22 | 27 | 1.43 | 0.83 | 2.48 |
| **Women** | | | | | | | | | | | | | | | | |
| Sweden | 6436 | 1 | | | 81018 | 1 | | | 332 | 1 | | | 17474 | 1 | | |
| Nordic countries | 415 | 1.01 | 0.87 | 1.17 | 4580 | 0.86 | 0.82 | 0.90 | 11 | 0.62 | 0.26 | 1.49 | 1197 | 1.02 | 0.93 | 1.11 |
| Southern Europe | 33 | 1.03 | 0.63 | 1.70 | 201 | 0.54 | 0.44 | 0.65 | 1 | 0.83 | 0.05 | 14.50 | 76 | 0.86 | 0.62 | 1.19 |
| Western Europe | 64 | 0.93 | 0.65 | 1.34 | 964 | 0.93 | 0.85 | 1.02 | 6 | 2.13 | 0.65 | 6.92 | 214 | 0.97 | 0.79 | 1.18 |
| Eastern Europe | 202 | **1.78** | **1.44** | **2.19** | 938 | 0.93 | 0.84 | 1.02 | 13 | **2.97** | **1.29** | **6.86** | 335 | **1.38** | **1.17** | **1.62** |
| Baltic countries | 10 | 0.88 | 0.36 | 2.17 | 191 | 0.85 | 0.69 | 1.05 | 0 | | | | 40 | 0.87 | 0.55 | 1.36 |
| Central Europe | 87 | 1.31 | 0.96 | 1.78 | 642 | 0.82 | 0.73 | 0.92 | 3 | 0.96 | 0.18 | 5.03 | 187 | 1.02 | 0.83 | 1.26 |
| Africa | 78 | **2.41** | **1.73** | **3.36** | 201 | 0.93 | 0.76 | 1.14 | 1 | 0.83 | 0.05 | 14.55 | 99 | **1.76** | **1.32** | **2.36** |
| North America | 10 | 0.68 | 0.27 | 1.67 | 134 | 0.74 | 0.58 | 0.94 | 1 | 1.72 | 0.10 | 30.05 | 27 | 0.67 | 0.39 | 1.17 |
| Latin America | 58 | 1.42 | 0.97 | 2.07 | 330 | 0.91 | 0.78 | 1.07 | 5 | 2.60 | 0.71 | 9.48 | 105 | 1.16 | 0.87 | 1.53 |
| Asia | 473 | **2.40** | **2.07** | **2.78** | 1719 | **1.14** | **1.06** | **1.23** | 12 | 1.38 | 0.58 | 3.29 | 670 | **1.80** | **1.60** | **2.02** |
| Russia | 16 | 1.17 | 0.57 | 2.39 | 113 | 0.79 | 0.61 | 1.04 | 0 | | | | 48 | **1.51** | **1.00** | **2.28** |

\*: Fully adjusted.

Bold values are statistically significant.

not the case among immigrant women. However, several immigrant subgroups showed a higher risk, especially men and women from Asia and women from Africa, and also, to a moderate extent, men and women from Eastern Europe. This was partly in-line with our hypothesis, i.e., that there is an increased risk of hearing impairment among certain immigrant groups to Sweden. For the different subgroups of hearing impairment, the pattern with a higher risk in immigrants from Eastern Europe, Africa and Asia seemed to be strongest for conductive hearing impairment.

The lower risk in many immigrant groups, i.e. those from North America and many European countries, except for immigrants from Eastern Europe, might to some extent represent a "healthy migrant effect", meaning that migrating individuals tend to have particularly good health compared to individuals living in the same country of origin [16]. However, we have no detailed information to support this hypothesis, and an earlier Danish study recommended using this theory of "healthy migrant effect with caution" [17]. Another potential explanation for the lower rates may be that immigrants with poor health and hearing loss may move back,

to a higher degree, to their home countries, than Swedish-born individuals with the same health problems, a phenomenon called salmon bias.

The increased risk for hearing impairment found in men and women from some regions, i.e. Asia, and in women from Africa, was found for both conductive hearing loss and other types of hearing impairment. These findings would need further studies of potential differences in risk factors between different ethnic groups, such as differences in genetic risk factors and relevant environmental exposures, e.g., noise. In an earlier study conducted by our group, we found than an increased risk in boys with parents from Asia could indicate that hereditary factors could be of importance [11]. The finding of an increased risk among men and women from Bosnia is also of interest. Regarding immigrants from Bosnia, they are refugees to a higher extent than other immigrants, as also immigrants from Iran and Iraq are [18], and this is also of interest to study further. However, the risk was also higher among men and women from Lebanon and Turkey. Our findings are also in contrast to earlier findings with lower risks of hearing impairment in Middle Eastern countries [19]. There may be a higher risk among refugees from war-torn countries where individuals may have been exposed to noise from gunfire and bombing. Untreated infections due to poorer health care in the country of origin, problems obtaining proper health care during the migration process, and differences in noise exposure in different countries may also explain the differences.

Regarding sex patterns, we found a slightly higher incidence for men compared to women, i.e. 4.7 vs 4.2%, which is in-line with findings in the rest of the world [2]. However, the excess risk among men was lower than that in other studies, e.g., as shown in an American study including adults aged ≥18 years of age that found a 60% higher risk of hearing loss among men than what was found in women [3]. We also found that being married was associated with a higher risk for hearing impairment among men. Married men could be more prone to seek health care and be diagnosed with a hearing impairment, as their medical problems could be noticed by their spouses. For example, we have previously noticed a lower risk of being diagnosed with dementia in unmarried and widowed men with atrial fibrillation in contrast to what could be expected, which could be interpreted that the disease is diagnosed at later stages [20]. Women have been found to seek health care more often than men [21].

Among socioeconomic factors, we found that a higher educational level was associated with a higher risk of hearing impairment, which is in contrast to an earlier American study [3]. One explanation behind these findings could be that health care seeking patterns might differ between different socioeconomic groups, where highly educated individuals may be more likely to seek care for their health problems than individuals with low education. However, a review of access to rehabilitation services, including hearing rehabilitation, concluded that no clear patterns could be seen as regards different factors including socioeconomic status, mostly owing to a lack of studies [22], although hearing loss is often underdiagnosed in the general population [23].

The co-morbidity patterns did not differ much between Swedish-born and foreign-born individuals, most likely owing to the fact that many of the foreign-born individuals are of European origin, with similar disease patterns as in Sweden. Tinnitus is of special interest, as it is highly associated with hearing impairment [24] and could be worthwhile to analyze separately. The higher risk of having both hearing and visual impairment, even if the excess risk was small, is also of special interest, i.e., the so-called dual-sensory impairment [4]. In individuals with the dual-sensory impairment, their health is even more affected [25] and also associated with a higher risk of early mortality [26, 27]. One study, which was conducted among individuals aged 55 years and older, found a higher risk of falls among individuals with dual-sensory impairment. However, that disappeared when also adjusting for cognitive impairment [28]. As regards falls, we included intracranial trauma as an indicator of more serious falls.

Some other causes of hearing impairment, such as infections and ototoxicity from certain drugs, are difficult to capture in registers [29] and beyond the scope of the present study. In women, dementia was associated with a lower risk of hearing impairment. This might be due to hearing loss being underdiagnosed in individuals with dementia, due to difficulties in performing hearing tests and difficulties trying out, learning to use, and managing a hearing aid.

Identifying hearing impairment is important as correction with hearing aids and rehabilitative services, such as auditory and communication training, have been shown to increase well-being and quality of life [9], especially among immigrants, who otherwise will have even greater difficulties learning a new language.

There are some limitations of this study. We used data from the NPR, where cases of deafness are likely to be identified, whereas other degrees of hearing impairment could be missed. Systemic errors could be problematic when using registry databases, and it is difficult to check the accuracy of the diagnoses. In addition, since primary health care data were not used, it is possible that some individuals with hearing impairment could have been missed. The number of cases diagnosed as presbycusis was low, most likely to be underestimated. Furthermore, the number of cases of hearing loss was low in some immigrant groups, especially when categorized into subgroups of hearing impairment. The care seeking patterns could differ between different groups of immigrants and also between immigrants and Swedish-born individuals, which also could have affected our results. Other individual characteristics, that may differ between immigrant groups and Swedish-born individuals, such as trust in physicians and chronic conditions not included in our data registers, could have influenced our results [15].

Finally, we did not have access to more specific data on the levels of hearing impairment among the individuals in Sweden.

There are also certain strengths of the present study. For example, the overall quality of the Swedish registers is regarded to be high, both as regards the Total National Population Register [12], and the NPR [30]. Furthermore, as used in the present study, the Swedish personal identification number allows for linkages between different Swedish registers [13]. All data were analyzed using pseudonymized serial numbers to secure all individuals' integrity.

In conclusion, we found an overall somewhat lower risk of hearing impairment in immigrant men but not in immigrant women. The findings of higher risks in some immigrant groups may need further attention because hearing loss may affect learning of the Swedish language and other aspects of successful integration. This suggests a need of an increased clinical awareness of potential hearing impairment when encountering patients belonging to certain immigrant groups.

## Supporting information

**S1 File.**
(DOCX)

## Acknowledgments

We thank Patrick Reilly for language editing.

## Author Contributions

**Conceptualization:** Axel Carlsson, Jan Sundquist, Kristina Sundquist.

**Data curation:** Xinjun Li.

**Funding acquisition:** Kristina Sundquist.

**Methodology:** Per Wändell, Xinjun Li, Axel Carlsson.

**Writing – original draft:** Per Wändell.

**Writing – review & editing:** Xinjun Li, Axel Carlsson, Jan Sundquist, Kristina Sundquist.

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
