## [Decision Letter · Decision Letter 0]

14 Mar 2022

PONE-D-22-03415Hearing impairment among adult foreign-born and Swedish-born individuals: a national Swedish studyPLOS ONE

Dear Dr. Carlsson,

Thank you for submitting your manuscript to PLOS ONE. After careful consideration, we feel that it has merit but does not fully meet PLOS ONE’s publication criteria as it currently stands. Therefore, we invite you to submit a revised version of the manuscript that addresses the points raised during the review process.

In revising your manuscript, please pay particular attention to the reviewers' suggestions for improving the rationale for the study and the reporting of research results.  Please include a point-by-point response to the reviewers' comments in your resubmission.

We look forward to receiving your revised manuscript.

Kind regards,

Jeffrey Jutai

Academic Editor

PLOS ONE

Journal Requirements:

Additional Editor Comments:

Thank you for selecting this journal for your manuscript submission. Your paper has received very thorough and constructive evaluations that indicate that it would require major revisions before being accepted for publication. Should you decide to submit a revised manuscript, please include a point-by-point response to the reviewers' concerns.

Reviewers' comments:

Reviewer's Responses to Questions

**Comments to the Author**

1. Is the manuscript technically sound, and do the data support the conclusions?

Reviewer #1: Partly

Reviewer #2: Partly

2. Has the statistical analysis been performed appropriately and rigorously? 

Reviewer #1: Yes

Reviewer #2: Yes

3. Have the authors made all data underlying the findings in their manuscript fully available?

Reviewer #1: Yes

Reviewer #2: No

4. Is the manuscript presented in an intelligible fashion and written in standard English?

Reviewer #1: Yes

Reviewer #2: Yes

5. Review Comments to the Author

Reviewer #1: Regarding the revision to the manuscript.

One concern is about using a measure of hearing impairment that is based on having a registered diagnosis. In the discussion you allude to the fact that there are not likely nativity differences in seeking behaviors, but there is no evidence or citation provided. Are the foreign born in Sweden less likely to have a routine source of care or to seek care when it is needed? How might language ability inform who receives a registered diagnosis?

Please clarify the timing of receiving a hearing impairment diagnosis in the text. After reading the paper, it is clear that the diagnosis happens in Sweden. Could immigrants who received a diagnosis, in their home country, prior to migrating be missed in the NPR? Is everyone in the NPR evaluated for a hearing impairment?

What is the substantive or analytic value at analyzing hearing impairments separately by types? You mention small number of cases for some types of hearing impairments, how small? In Table 2 are all hearing impairments combined together? When you refer to hearing impairment generally in the discussion when summarizing results, what type or types of impairments are included?

What if any are the limitations of not having patient diagnoses of hearing impairments from primary health care?

Did you try models with just 2001 data and beyond from outpatient diagnoses? Are there differences between hospital and outpatient diagnoses?

Is it possible to include other co morbidities such as physical limitations?

Do you have other immigration related measures?

Here I am specifically thinking about factors like duration of residence in Sweden or age at arrival in Sweden? Cohort of arrival?

In addition to education, can you control for their language ability?

Do you know the age at which they were diagnosed with a hearing impairment?

The inclusion of the neighborhood SES is not clear. It is not theorized about in the front end of the paper. What is the importance on including? The models are set up in a way to examine whether/how SES explains some of the country of birth and hearing impairment association but again this is not theorized in the front end.

The conclusion needs revisions/expansion. Specifically, why does understanding hearing impairment by nativity matter in Sweden? Expand on this conclusion point about hearing difficulty and learning a language. What are the implications for integration for themselves or their children? Or for policy?

Reviewer #2: Thank you for allowing me to review this paper. While this study was conducted rather comprehensively, there are several major concerns with the justification for the study and written methodology/analytic approach.

The justification to conduct this study needs to be expanded upon- why is it important to better understand differences in risk between foreign-born and Swedish-born adults? Please consider highlighting this earlier in the introduction and expanding on the decision to present only sex stratified results and the benefits for this approach. The discussion would also benefit from expanding on the implications of study findings for the target population.

The relevance of associations with comorbidities is unclear throughout the manuscript. How were these factors chosen (e.g., via existing literature and/or by evaluating confounding in the data; this needs to be expanded upon in introduction & methods and more citations are needed that are specific to hearing loss)? It is not clear why region is divided into urban, northern, and southern Sweden- does urbanicity vary across northern/southern Sweden? The NSES is built with both an education status variable and a geographical variable, although different measures of education and geography were already adjusted for. This introduces substantial concerns of collinearity of the covariates included in Models 2 and 3.

A strength of this study is that it uses longitudinal data to evaluate relative risk of hearing impairment. However, there is a lack of detail on the timeline of the study period which makes it difficult to interpret results. When were baseline characteristics measured, was it before Jan 1, 1998? Please provide more information of longitudinal follow up in this sample and how different times to the event were handled. How were patients treated that died or emigrated prior to the end of the study period? What was the average time to event?

Methods 2.1 states that codes were assigned when treated. Is this true, or were codes assigned when treated or diagnosed? The division of types of hearing loss (i.e., conductive, sensorineural, other) described in the methods section does not match how data are presented in Table 3. What does ‘other’ refer to in the methods?

What was the motivation for stratifying to categories conductive and ‘other’ (Table 3)? The ‘other’ category in Table 3 appears to include several types of hearing loss (i.e., sensorineural, mixed, ototoxic, other causes; this also needs to be labeled on the table) that would likely have different etiologies, and for which the risk would vary across foreign- and Swedish-born for several different reasons. The broad categorization of the ‘other’ category may also mask relevant associations. Is adjustment for the covariates in Model 3 relevant for these different types of hearing loss given their different etiologies?

The discussion highlights that risk of hearing impairment in foreign-born men is lower than Swedish-born men. Please highlight that the risk is low (HR = 0.97) and discuss why the overall HR is low (although HRs appear substantially different after further stratification).

The limitations section should mention systematic errors present in registry databases. For example, the methods states that codes are only available from specialists, not primary care – please discuss how that would impact study results. Hearing loss is often underdiagnosed despite it being present – please discuss how this limitation of using registry data impacts results. The strengths section states there are no anticipated differences in foreign and Swedish individuals in health care seeking because access is good. Access and health care seeking are different concepts and health care seeking can be motivated by several factors that are not related to only access. Please clarify. Please expand on factors that were potentially uncontrolled for in these analyses.

Lastly, there are numerous grammatical errors and incorrect word choices throughout the manuscript. It is recommended that authors carefully proofread the manuscript prior to resubmission.

Specific comments:

Introduction:

P1, sentence 2: please rephrase. Etiologies and risk should be 2 separate concepts.

P1, last sentence: what was the age range of the study reporting 6% hearing loss prevalence? That study is from 1992, there are numerous cohort studies with more recent data.

P5: unclear what ‘non-Western regions’ refers to. Most of the earlier text focuses on risk differences in Western Europe/United states vs Asian regions but the hypothesis states there will be anticipated differences also in Central/Eastern Europe. Please clarify.

There are several places where the word ‘cause’ is used in reference to comorbidities. Please reword. It is not biologically plausible that some of these comorbidities (e.g., visual impairment) cause hearing impairment.

Please avoid single-sentence paragraphs.

Results:

-Methods state that age is used continuously in models, but age is presented categorically in tables. Please also provide mean / SD / range of age in text.

-Please consider reorganizing the results section to include headers on the topic being presented. Please consider including %s rather than only numerators/denominators in the text and tables. In several sections of the results, it is unclear which table is being referred to (e.g., ‘incidence rate’ in >60 and <=60 yrs).

-Considering commenting on protective HR for dementia association in women.

-Please consider reformatting tables to make it clearer when subcategories exist (e.g., Table 2: could indent regions under ‘all foreign born’ categories). Authors may also consider creating a header for ‘hospital diagnosis of…’ then listing conditions underneath.

-Table 1: please consider statistically evaluating differences in baseline characteristics between men and women.

-Prevalence of visual impairment seems to be very low. Please discuss why and potential implications for interpretations of results given that hearing and visual impairment often co-occur in aging.

-Please provide more detailed footnotes on adjustment for tables (e.g., Table 3, Supplementary tables)

Discussion:

-authors compare results to a prevalence study [3] showing 60% higher odds (not risk) of hearing loss in men vs women. Cross-sectional results from that study, presented as OR, cannot be directly compared to these longitudinal study results, presented as HR.

-It is not clear how the male-female health-survival paradox fits with study findings – please clarify. Citations are needed for statements on sex and socioeconomic differences in health care seeking.

-what does ‘disease patterns’ refer to? Please clarify.

-why would healthy migrant effects only apply to immigrants from certain regions?

6. PLOS authors have the option to publish the peer review history of their article (what does this mean?). If published, this will include your full peer review and any attached files.

Reviewer #1: No

Reviewer #2: No

---

## [Author Response · Author response to Decision Letter 0]

27 May 2022

Journal Requirements:

Our response: We have revised to the best of our abilities.

Our response: The provided funding information was not updated, and we apologize for that. We have now excluded the mentioned grants from the funding information and financial disclosure section as we found that they were not relevant for the present study. 

Our response: Because of ethical and legal restrictions in Sweden, the authors are not allowed to share the data. However, it is possible for researchers to apply for anonymized datasets from the Swedish authorities (i.e., the National Board of Health and Welfare and Statistics Sweden). 

Our response: The ethics statement is now included in the Methods section. 

Additional Editor Comments:

Thank you for selecting this journal for your manuscript submission. Your paper has received very thorough and constructive evaluations that indicate that it would require major revisions before being accepted for publication. Should you decide to submit a revised manuscript, please include a point-by-point response to the reviewers' concerns.

Reviewers' comments:

Reviewer's Responses to Questions

Comments to the Author

1. Is the manuscript technically sound, and do the data support the conclusions?

Reviewer #1: Partly

Reviewer #2: Partly

2. Has the statistical analysis been performed appropriately and rigorously? 

Reviewer #1: Yes

Reviewer #2: Yes

3. Have the authors made all data underlying the findings in their manuscript fully available?

Reviewer #1: Yes

Reviewer #2: No

4. Is the manuscript presented in an intelligible fashion and written in standard English?

Reviewer #1: Yes

Reviewer #2: Yes

5. Review Comments to the Author

Reviewer #1: Regarding the revision to the manuscript.

One concern is about using a measure of hearing impairment that is based on having a registered diagnosis. In the discussion you allude to the fact that there are not likely nativity differences in seeking behaviors, but there is no evidence or citation provided. Are the foreign born in Sweden less likely to have a routine source of care or to seek care when it is needed? How might language ability inform who receives a registered diagnosis?

Our response: There are only a few studies regarding the context of equal care. The available studies have been undertaken on other patient groups with different diseases or diagnoses. In a study of second myocardial infarctions, low socioeconomic status was a risk factor for a less routine source of care, which remained after adjustments for immigrant status. Immigrants do not seem less likely to have a routine source of care. Language problems might of course affect the understanding of what a specific diagnosis means in practice and daily life, but we have not found studies to support that claim. If requested, we could delete that statement from the discussion section.

Please clarify the timing of receiving a hearing impairment diagnosis in the text. After reading the paper, it is clear that the diagnosis happens in Sweden. Could immigrants who received a diagnosis, in their home country, prior to migrating be missed in the NPR? Is everyone in the NPR evaluated for a hearing impairment?

Our response: Surely an immigrant could have received a diagnosis for hearing impairment in the country of origin. However, to get access to hearing aids, which are subsidized in Sweden, a visit to the Swedish healthcare system is needed, hence why we expect that the risk of being missed in the NPR should be small. Everyone will not be evaluated as the NPR is based upon a diagnosis after a clinical visit.

What is the substantive or analytic value at analyzing hearing impairments separately by types? You mention small number of cases for some types of hearing impairments, how small? In Table 2 are all hearing impairments combined together? When you refer to hearing impairment generally in the discussion when summarizing results, what type or types of impairments are included?

Our response: We have separated hearing impairments by types as the causes behind them may vary. We have now added the numbers of the specific diagnoses for men and women (see the new Supplementary Tables 6a and 6b). 

What if any are the limitations of not having patient diagnoses of hearing impairments from primary health care?

Our response: Hearing impairment, although often suspected in primary care, should normally be diagnosed in secondary care, hence why we could expect that the limitations of not having primary care diagnoses should be limited.

Did you try models with just 2001 data and beyond from outpatient diagnoses? Are there differences between hospital and outpatient diagnoses?

Our response: We have not performed such analyses as we believed that the main question is that the diagnoses are captured, whether in inpatients or outpatients. However, if this is requested we are willing to conduct such analyses.

Is it possible to include other co morbidities such as physical limitations?

Our response: We included important diagnoses, such as stroke and arthropathy, in order to capture physical limitations. Otherwise, it may be difficult to catch other conditions leading to physical limitations as many of these conditions are cared for in primary care.

Do you have other immigration related measures?

Here I am specifically thinking about factors like duration of residence in Sweden or age at arrival in Sweden? Cohort of arrival?

Our response: We have now added a sensitivity analysis with adjustment for duration of residence in Sweden/time in Sweden (see the new Supplementary Table 5).

In addition to education, can you control for their language ability?

Our response: No, we have no data on this.

Do you know the age at which they were diagnosed with a hearing impairment?

Our response: Yes, we have the first event and age registered.

The inclusion of the neighborhood SES is not clear. It is not theorized about in the front end of the paper. What is the importance on including? The models are set up in a way to examine whether/how SES explains some of the country of birth and hearing impairment association but again this is not theorized in the front end.

Our response: Neighborhood SES has been shown to be an important socioeconomic factor for several health outcomes and we have added a sentence to justify its inclusion (inserted on p. 6). 

The conclusion needs revisions/expansion. Specifically, why does understanding hearing impairment by nativity matter in Sweden? Expand on this conclusion point about hearing difficulty and learning a language. What are the implications for integration for themselves or their children? Or for policy?

Our response: For planning actions in the health care system and society, it is important to know patterns of hearing impairment in different immigrant groups. Furthermore, hearing impairment may affect integration in Sweden as it potentially will increase the language barriers.

Reviewer #2: Thank you for allowing me to review this paper. While this study was conducted rather comprehensively, there are several major concerns with the justification for the study and written methodology/analytic approach.

The justification to conduct this study needs to be expanded upon- why is it important to better understand differences in risk between foreign-born and Swedish-born adults? Please consider highlighting this earlier in the introduction and expanding on the decision to present only sex stratified results and the benefits for this approach. The discussion would also benefit from expanding on the implications of study findings for the target population.

Our response: To know patterns of hearing impairment in different immigrant groups is of importance for both the health care system and to society in general; this is for planning actions and to identify hearing impairment early on after migration in order to Sweden to give immigrants the best possible opportunity to learn Swedish and to get integrated in Sweden. Investigating potential differences by sex is also important as men and women have different life circumstances worldwide.

The relevance of associations with comorbidities is unclear throughout the manuscript. How were these factors chosen (e.g., via existing literature and/or by evaluating confounding in the data; this needs to be expanded upon in introduction & methods and more citations are needed that are specific to hearing loss)? It is not clear why region is divided into urban, northern, and southern Sweden- does urbanicity vary across northern/southern Sweden? The NSES is built with both an education status variable and a geographical variable, although different measures of education and geography were already adjusted for. This introduces substantial concerns of collinearity of the covariates included in Models 2 and 3. 

Our response: We included the selected comorbidities as several of them are related to physical limitations which often are seen together with hearing impairment. The rough geographical pattern was chosen to adjust for possible differences in health care between the more densely populated areas in the larger cities and southern Sweden, and the less densely populated areas in northern Sweden. The neighborhood SES includes different factors, where low educational status is only one. It is true that there is some overlapping between the different covariates. However, we don’t entirely agree that large problems with collinearity exist as the different SES factors reflect different assessments at different levels. We have previously shown, for example, that neighborhood SES is a risk factor for appropriate prescribed medications, stroke and myocardial infarction that is independent of individual education level and migration status. 

A strength of this study is that it uses longitudinal data to evaluate relative risk of hearing impairment. However, there is a lack of detail on the timeline of the study period which makes it difficult to interpret results. When were baseline characteristics measured, was it before Jan 1, 1998? Please provide more information of longitudinal follow up in this sample and how different times to the event were handled. How were patients treated that died or emigrated prior to the end of the study period? What was the average time to event? 

Our response: We used an open cohort design and as Cox regression was used in the statistical analysis, only the first event was registered. Risk time was calculated until the event and those who died or emigrated were censored (these sentences are now included on pp. 6-7). We have now clarified this in the methods section. We have now also included mean time to event (Results, p. 8).

Methods 2.1 states that codes were assigned when treated. Is this true, or were codes assigned when treated or diagnosed? The division of types of hearing loss (i.e., conductive, sensorineural, other) described in the methods section does not match how data are presented in Table 3. What does ‘other’ refer to in the methods?

Our response: The codes refer to clinical diagnoses and the codes were given on the first occasion. We chose to stratify into the categories “Conductive and sensorineural hearing loss” and “Other hearing loss”, and have clarified this in the methods. Thus, the texts in Methods and Table 3 are now congruent. We have included the specific diagnoses and codes in supplementary Table 6a. Data on treatment are not included in the present manuscript. 

What was the motivation for stratifying to categories conductive and ‘other’ (Table 3)? The ‘other’ category in Table 3 appears to include several types of hearing loss (i.e., sensorineural, mixed, ototoxic, other causes; this also needs to be labeled on the table) that would likely have different etiologies, and for which the risk would vary across foreign- and Swedish-born for several different reasons. The broad categorization of the ‘other’ category may also mask relevant associations. Is adjustment for the covariates in Model 3 relevant for these different types of hearing loss given their different etiologies?

Our response: We chose to stratify into the categories “Conductive and sensorineural hearing loss” and “Other hearing loss”, and have clarified this in the methods. Thus, the texts in Methods and Table 3 are now congruent. We have included all diagnoses in Supplementary Table 6a. We agree that the choice of covariates could be questioned, but the adjustment for co-morbidities only changed the HRs marginally. 

The discussion highlights that risk of hearing impairment in foreign-born men is lower than Swedish-born men. Please highlight that the risk is low (HR = 0.97) and discuss why the overall HR is low (although HRs appear substantially different after further stratification).

Our response: We have extended the discussion; several groups showed a lower risk, i.e. men and women from the Nordic countries, Southern Europe, and North America. Individuals from the Nordic countries constitute a large group in Sweden, and the lower risk in the mentioned groups explains why the risk is somewhat lower overall. In addition, there is the possibility of healthy migrant effects, i.e. that individuals with poorer health, including hearing impairment, are less likely to migrate.

The limitations section should mention systematic errors present in registry databases. For example, the methods states that codes are only available from specialists, not primary care – please discuss how that would impact study results. Hearing loss is often underdiagnosed despite it being present – please discuss how this limitation of using registry data impacts results. The strengths section states there are no anticipated differences in foreign and Swedish individuals in health care seeking because access is good. Access and health care seeking are different concepts and health care seeking can be motivated by several factors that are not related to only access. Please clarify. Please expand on factors that were potentially uncontrolled for in these analyses.

Our response: We have extended the discussion, accordingly.

Lastly, there are numerous grammatical errors and incorrect word choices throughout the manuscript. It is recommended that authors carefully proofread the manuscript prior to resubmission.

Our response: The manuscript has now been proofread by a native English-speaking science editor. 

Specific comments:

Introduction:

P1, sentence 2: please rephrase. Etiologies and risk should be 2 separate concepts.

Our response: We have now reworded the sentence and divided it into two sentences. 

P1, last sentence: what was the age range of the study reporting 6% hearing loss prevalence? That study is from 1992, there are numerous cohort studies with more recent data.

Our response: Reference article 3 by Li et al was published in 2018, and describes two surveys from 2014 and 2016, respectively, and the sub-studies include individuals 18 years of age and above. The article from 1992 was authored by Winkleby et al concerning socioeconomic status.

P5: unclear what ‘non-Western regions’ refers to. Most of the earlier text focuses on risk differences in Western Europe/United states vs Asian regions but the hypothesis states there will be anticipated differences also in Central/Eastern Europe. Please clarify.

Our response: We agree that the wording of “Western” and “non-Western” regions is often vague and have omitted these categories in the manuscript.

There are several places where the word ‘cause’ is used in reference to comorbidities. Please reword. It is not biologically plausible that some of these comorbidities (e.g., visual impairment) cause hearing impairment.

Our response: We agree and have changed the wording as suggested.

Please avoid single-sentence paragraphs.

Our response: We have avoided this now.

Results:

-Methods state that age is used continuously in models, but age is presented categorically in tables. Please also provide mean / SD / range of age in text.

Our response: We have now added mean age + SD in the text.

-Please consider reorganizing the results section to include headers on the topic being presented. 

Our response: We have now included headers.

Please consider including %s rather than only numerators/denominators in the text and tables. In several sections of the results, it is unclear which table is being referred to (e.g., ‘incidence rate’ in >60 and <=60 yrs).

Our response: We have now tried to specify better which tables are being referred to. We prefer to keep both numbers and percentages in Table 1 and the Supplementary Tables 1a and 1b, but could change this if it is requested.

-Considering commenting on protective HR for dementia association in women.

Our response: We have now added a sentence on this.

Discussion, p. 12: “In women, dementia was associated with a lower risk of hearing impairment. This might be due to hearing loss being underdiagnosed in individuals with dementia, due to difficulties in performing hearing tests and difficulties trying out, learning to use, and managing a hearing aid.” 

-Please consider reformatting tables to make it clearer when subcategories exist (e.g., Table 2: could indent regions under ‘all foreign born’ categories). Authors may also consider creating a header for ‘hospital diagnosis of…’ then listing conditions underneath.

-Table 1: please consider statistically evaluating differences in baseline characteristics between men and women.

Our response: We have changed the comorbidity sections in Table 2 and included “hospital diagnosis” as a header. Regarding the differences between men and women in general we prefer not to evaluate the differences, as the main theme is difference between foreign-born and Swedish-born individuals. We do believe that the difference in percentages between men and women could easily be found for the interested reader.

-Prevalence of visual impairment seems to be very low. Please discuss why and potential implications for interpretations of results given that hearing and visual impairment often co-occur in aging.

Our response: We have no confident explanation for this. One possible explanation is that low or moderate degrees of visual impairment among elderly individuals are not noticed and registered.

-Please provide more detailed footnotes on adjustment for tables (e.g., Table 3, Supplementary tables)

Our response: We have done so. 

Discussion:

-authors compare results to a prevalence study [3] showing 60% higher odds (not risk) of hearing loss in men vs women. Cross-sectional results from that study, presented as OR, cannot be directly compared to these longitudinal study results, presented as HR.

Our response: The results in that study present weighted prevalence and adjusted prevalence rates (PRs), hence why we think that the wording is correct. We have, however, added a sentence that these results may not be directly comparable with ours.

-It is not clear how the male-female health-survival paradox fits with study findings – please clarify.

Our response: This has mostly been discussed in relation to heart diseases, and we agree that it is difficult to discuss this term here, hence why we have omitted it.

Citations are needed for statements on sex and socioeconomic differences in health care seeking.

Our response: We have included a systematic review on access to rehabilitation services, where the conclusion is that “no clear patterns were seen in access by equity measures such as age, locality, socioeconomic status, or country income group”, which is due to a low number of studies. Seeking patterns could be more complex, however, and it is difficult to generalize. In some studies, we have previously found some results that could be because some factors could be associated with earlier care seeking behaviors, mostly for married people whereas unmarried individuals without a partner seem to react later, e.g. for dementia or heart failure. However, we have no data supporting this on hearing impairment.

-what does ‘disease patterns’ refer to? Please clarify.

Our response: We agree that this is vaguely expressed and have changed the wording accordingly.

-why would healthy migrant effects only apply to immigrants from certain regions?

Our response: As regards to the healthy migrant effect there is no detailed information on this for different groups; an earlier Danish study recommended to use this theory with caution. However, in some of the immigrant groups with lower risk than among Swedish-born individuals, we could have expected a similar rather than a lower risk and this could be related to a healthy migrant effect. 

6. PLOS authors have the option to publish the peer review history of their article (what does this mean?). If published, this will include your full peer review and any attached files.

Do you want your identity to be public for this peer review? For information about this choice, including consent withdrawal, please see our Privacy Policy.

Reviewer #1: No

Reviewer #2: No

---

## [Decision Letter · Decision Letter 1]

1 Jul 2022

PONE-D-22-03415R1Hearing impairment among adult foreign-born and Swedish-born individuals: a national Swedish studyPLOS ONE

Dear Dr. Carlsson,

Thank you for submitting your manuscript to PLOS ONE. After careful consideration, we feel that it has merit but does not fully meet PLOS ONE’s publication criteria as it currently stands. Therefore, we invite you to submit a revised version of the manuscript that addresses the points raised during the review process.

The author is requested to respond to the comments from Reviewer #2 before the manuscript can be considered for publication. 

We look forward to receiving your revised manuscript.

Kind regards,

Jeffrey Jutai

Academic Editor

PLOS ONE

Additional Editor Comments:

Further revision is required before the manuscript can be considered for publication. The authors should revise the manuscript in response to the comments from Reviewer #2.

Reviewers' comments:

Reviewer's Responses to Questions

**Comments to the Author**

1. If the authors have adequately addressed your comments raised in a previous round of review and you feel that this manuscript is now acceptable for publication, you may indicate that here to bypass the “Comments to the Author” section, enter your conflict of interest statement in the “Confidential to Editor” section, and submit your "Accept" recommendation.

Reviewer #1: All comments have been addressed

Reviewer #2: (No Response)

2. Is the manuscript technically sound, and do the data support the conclusions?

Reviewer #1: Yes

Reviewer #2: Partly

3. Has the statistical analysis been performed appropriately and rigorously? 

Reviewer #1: Yes

Reviewer #2: Yes

4. Have the authors made all data underlying the findings in their manuscript fully available?

Reviewer #1: (No Response)

Reviewer #2: No

5. Is the manuscript presented in an intelligible fashion and written in standard English?

Reviewer #1: Yes

Reviewer #2: Yes

6. Review Comments to the Author

Reviewer #1: (No Response)

Reviewer #2: Thank you for allowing me to review the revised draft of this paper. Below are comments based on these revisions. There are several points that remain unaddressed in the paper. The most important remaining concern is how hearing loss is classified and presented in Table 3.

Previous comment: The justification to conduct this study needs to be expanded upon- why is it important to better understand differences in risk between foreign-born and Swedish-born adults? Please consider highlighting this earlier in the introduction and expanding on the decision to present only sex stratified results and the benefits for this approach. The discussion would also benefit from expanding on the implications of study findings for the target population. ****Authors chose not to modify the text although these minor revisions may be helpful to the reader. Please reconsider.

Previous comment: The relevance of associations with comorbidities is unclear throughout the manuscript. How were these factors chosen (e.g., via existing literature and/or by evaluating confounding in the data; this needs to be expanded upon in introduction & methods and more citations are needed that are specific to hearing loss)? It is not clear why region is divided into urban, northern, and southern Sweden- does urbanicity vary across northern/southern Sweden? ****Authors chose not to modify the text although these minor revisions may be helpful to the reader. Please reconsider.

There remains a substantial issue (highlighted in previous comments) regarding how hearing loss is classified in Table 3. It is not clear why sensorineural hearing loss and conductive hearing loss are grouped together. Furthermore, diagnoses that present as sensorineural hearing loss (e.g., presbycusis, most types of ototoxic hearing loss) are misclassified in the ‘other’ category. Again, sensorineural and conductive hearing losses often have drastically different etiologies and there are regional differences in these etiologies. For example, permanent conductive hearing loss could more likely be influenced by differences in health care across countries (e.g., untreated ear infections) and is more highly prevalent in countries with lower income. Permanent sensorineural hearing loss is more likely due to aging/related processes and environmental and health-related risk factors. Again, there are different distributions of relevant exposures/risk factors across different regions/countries. I would strongly urge authors to re consider how these groups are defined. The line added to the discussion about presbycusis being underdiagnosed may not be true, as presbycusis may instead be diagnosed as sensorineural hearing loss. It is known that hearing loss, in general, is often underdiagnosed given patients’ underreporting of hearing difficulties and because many providers do not prioritize detection of hearing loss or appropriate referrals until hearing loss is more severe. This point, mentioned previously, also remains unaddressed in this revision.

Previous comment: The limitations section should mention systematic errors present in registry databases. For example, the methods states that codes are only available from specialists, not primary care – please discuss how that would impact study results. Hearing loss is often underdiagnosed despite it being present – please discuss how this limitation of using registry data impacts results. The strengths section states there are no anticipated differences in foreign and Swedish individuals in health care seeking because access is good. Access and health care seeking are different concepts and health care seeking can be motivated by several factors that are not related to only access. Please clarify. Please expand on factors that were potentially uncontrolled for in these analyses. ****Most of these points remain unaddressed. Please reconsider.

Previous comment: P1, last sentence: what was the age range of the study reporting 6% hearing loss prevalence? ****Information on the age range used to determine prevalence should be included in the text. Please reconsider.

Previous comment: In several sections of the results, it is unclear which table is being referred to (e.g., ‘incidence rate’ in >60 and <=60 yrs). ****No changes to address this are seen in the manuscript. Please reconsider.

Previous comment: what does ‘disease patterns’ refer to? Please clarify. ****Authors say the wording has been changed but it remains in the manuscript. Please reconsider.

7. PLOS authors have the option to publish the peer review history of their article (what does this mean?). If published, this will include your full peer review and any attached files.

Reviewer #1: No

Reviewer #2: No

---

## [Author Response · Author response to Decision Letter 1]

4 Aug 2022

Additional Editor Comments:

Further revision is required before the manuscript can be considered for publication. The authors should revise the manuscript in response to the comments from Reviewer #2.

Reviewers' comments:

Reviewer's Responses to Questions

Comments to the Author

1. If the authors have adequately addressed your comments raised in a previous round of review and you feel that this manuscript is now acceptable for publication, you may indicate that here to bypass the “Comments to the Author” section, enter your conflict of interest statement in the “Confidential to Editor” section, and submit your "Accept" recommendation.

Reviewer #1: All comments have been addressed

Reviewer #2: (No Response)

2. Is the manuscript technically sound, and do the data support the conclusions?

Reviewer #1: Yes

Reviewer #2: Partly

3. Has the statistical analysis been performed appropriately and rigorously? 

Reviewer #1: Yes

Reviewer #2: Yes

4. Have the authors made all data underlying the findings in their manuscript fully available?

Reviewer #1: (No Response)

Reviewer #2: No

5. Is the manuscript presented in an intelligible fashion and written in standard English?

Reviewer #1: Yes

Reviewer #2: Yes

6. Review Comments to the Author

Reviewer #1: (No Response)

Reviewer #2: Thank you for allowing me to review the revised draft of this paper. Below are comments based on these revisions. There are several points that remain unaddressed in the paper. The most important remaining concern is how hearing loss is classified and presented in Table 3.

Previous comment: The justification to conduct this study needs to be expanded upon- why is it important to better understand differences in risk between foreign-born and Swedish-born adults? Please consider highlighting this earlier in the introduction and expanding on the decision to present only sex stratified results and the benefits for this approach. The discussion would also benefit from expanding on the implications of study findings for the target population. ****Authors chose not to modify the text although these minor revisions may be helpful to the reader. Please reconsider.

Our response: Thank you for mentioning these issues again. We have now expanded on the justification to conduct this study. The acculturation (e.g., learning the new language) into the new homeland for foreign-born citizens is certainly depending on good hearing. Hearing impairment is therefore an important issue to study in order to provide help with correction, if possible. We have revised the following part of the introduction:

Introduction, p. 3: “As regards immigrants, hearing impairment may be challenging when learning the language and culture of a new country, which calls for attention and motivated us to conduct this study. Identifying hearing impairment is also important as correction with hearing aids and rehabilitative services have been shown to increase well-being and quality of life.” 

We have also added the following new text in the methods section to justify the stratification by sex:

“The analyses were stratified by sex because men and women experience different types of environments, including those related to occupation, and also have different health care-seeking patterns. https://bmcprimcare.biomedcentral.com/articles/10.1186/s12875-016-0440-0”

Please also note that there were differences in the results based on sex. The discussion mentions these findings in the first two sentences.

The following new sentence has also been added as a last sentence to the discussion section in order to better reflect the implications of the study:

“This suggests a need of an increased clinical awareness of potential hearing impairment when encountering patients belonging to certain immigrant groups.”

In addition, the following text can be found in the discussion section, just before the limitations:

“Identifying hearing impairment is important as correction with hearing aids and rehabilitative services, such as auditory and communication training, have been shown to increase well-being and quality of life [9], especially among immigrants, who otherwise will have even greater difficulties learning a new language.” 

Previous comment: The relevance of associations with comorbidities is unclear throughout the manuscript. How were these factors chosen (e.g., via existing literature and/or by evaluating confounding in the data; this needs to be expanded upon in introduction & methods and more citations are needed that are specific to hearing loss)? It is not clear why region is divided into urban, northern, and southern Sweden- does urbanicity vary across northern/southern Sweden? ****Authors chose not to modify the text although these minor revisions may be helpful to the reader. Please reconsider.

Our response: The chosen co-morbidities were based on existing literature and we have now added this information, together with references, in the methods section. Regarding regions in Sweden, we have added the following new text in the methods section:

“Regarding regions in Sweden, the urbanicity differs with many sparsely populated parts in northern Sweden with, in many cases, long transportation routes and also poorer access to ophthalmologists.”

There remains a substantial issue (highlighted in previous comments) regarding how hearing loss is classified in Table 3. It is not clear why sensorineural hearing loss and conductive hearing loss are grouped together. Furthermore, diagnoses that present as sensorineural hearing loss (e.g., presbycusis, most types of ototoxic hearing loss) are misclassified in the ‘other’ category. Again, sensorineural and conductive hearing losses often have drastically different etiologies and there are regional differences in these etiologies. For example, permanent conductive hearing loss could more likely be influenced by differences in health care across countries (e.g., untreated ear infections) and is more highly prevalent in countries with lower income. Permanent sensorineural hearing loss is more likely due to aging/related processes and environmental and health-related risk factors. Again, there are different distributions of relevant exposures/risk factors across different regions/countries. I would strongly urge authors to re consider how these groups are defined. The line added to the discussion about presbycusis being underdiagnosed may not be true, as presbycusis may instead be diagnosed as sensorineural hearing loss. It is known that hearing loss, in general, is often underdiagnosed given patients’ underreporting of hearing difficulties and because many providers do not prioritize detection of hearing loss or appropriate referrals until hearing loss is more severe. This point, mentioned previously, also remains unaddressed in this revision.

Our response: We agree with this comment and have now performed a new categorization of the diagnoses of hearing impairment, accordingly; we believe that the new classification is more accurate than the earlier, and the new results are shown in the new Table 3. In addition, we added the following new text to the last sentence in the fifth paragraph of the discussion:

“…although hearing loss is often underdiagnosed in the general population https://jamanetwork.com/journals/jamanetworkopen/fullarticle/2769843.”

Previous comment: The limitations section should mention systematic errors present in registry databases. For example, the methods states that codes are only available from specialists, not primary care – please discuss how that would impact study results. Hearing loss is often underdiagnosed despite it being present – please discuss how this limitation of using registry data impacts results. The strengths section states there are no anticipated differences in foreign and Swedish individuals in health care seeking because access is good. Access and health care seeking are different concepts and health care seeking can be motivated by several factors that are not related to only access. Please clarify. Please expand on factors that were potentially uncontrolled for in these analyses. ****Most of these points remain unaddressed. Please reconsider.

Our response: We agree that the mentioned factors are important to be expanded in the limitations section and have done so: 

Discussion, p. 13: 

“Systemic errors could be problematic when using registry databases, and it is difficult to check the accuracy of the diagnoses. In addition, since primary health care data were not used, it is possible that some individuals with hearing impairment could have been missed.”

“The care seeking patterns could differ between different groups of immigrants and also between immigrants and Swedish-born individuals, which also could have affected our results. Other individual characteristics, that may differ between immigrant groups and Swedish-born individuals, such as trust in physicians and chronic conditions not included in our data registers, could have influenced our results (https://bmcprimcare.biomedcentral.com/articles/10.1186/s12875-016-0440-0).”

We have deleted the sentence that states that we do not expect any major differences in health care seeking patterns between immigrants and Swedish-born individuals.

Previous comment: P1, last sentence: what was the age range of the study reporting 6% hearing loss prevalence? ****Information on the age range used to determine prevalence should be included in the text. Please reconsider.

Our response: The age range in the US article by Li et al was adults aged ≥ 18 years. This is now mentioned in the text.

Previous comment: In several sections of the results, it is unclear which table is being referred to (e.g., ‘incidence rate’ in >60 and <=60 yrs). ****No changes to address this are seen in the manuscript. Please reconsider.

Our response: We apologize for being unclear. We have now added the text “data not shown” in parenthesis for the specific example. As we already have several Supplementary Tables, we did not include the specific results from this example. We also added information on which table(s) we refer to in other locations in the results section. 

Previous comment: what does ‘disease patterns’ refer to? Please clarify. ****Authors say the wording has been changed but it remains in the manuscript. Please reconsider.

Our response: We agree and have changed the wording to “co-morbidity patterns” in the beginning of that sentence:

“The co-morbidity patterns did not differ much between Swedish-born and foreign-born individuals, most likely owing to the fact that many of the foreign-born individuals are of European origin, with similar disease patterns as in Sweden.”

7. PLOS authors have the option to publish the peer review history of their article (what does this mean?). If published, this will include your full peer review and any attached files.

Do you want your identity to be public for this peer review? For information about this choice, including consent withdrawal, please see our Privacy Policy.

Reviewer #1: No

Reviewer #2: No

---

## [Editor Report · Decision Letter 2]

9 Aug 2022

Hearing impairment among adult foreign-born and Swedish-born individuals: a national Swedish study

PONE-D-22-03415R2

Dear Dr. Carlsson,

We’re pleased to inform you that your manuscript has been judged scientifically suitable for publication and will be formally accepted for publication once it meets all outstanding technical requirements.

Kind regards,

Jeffrey Jutai

Academic Editor

PLOS ONE
---

## [Editor Report · Acceptance letter]

11 Aug 2022

PONE-D-22-03415R2 

Hearing impairment among adult foreign-born and Swedish-born individuals: a national Swedish study 

Dear Dr. Carlsson:

I'm pleased to inform you that your manuscript has been deemed suitable for publication in PLOS ONE. Congratulations! Your manuscript is now with our production department. 

Kind regards, 

on behalf of

Dr. Jeffrey Jutai 

Academic Editor

PLOS ONE